# A four-season study quantifying the weekly external training loads during different between match microcycle lengths in professional rugby league

**James Parmley**[1]*, **Ben Jones**[1,2,3,4,5], **Tom Sawczuk**[1,2,6], **Dan Weaving**[1,3]

**1** Carnegie Applied Rugby Research (CARR) Centre, Carnegie School of Sport, Leeds Beckett University, Leeds, United Kingdom, **2** England Performance Unit, Rugby Football League, Leeds, United Kingdom, **3** Leeds Rhinos Rugby League Club, Leeds, United Kingdom, **4** School of Science and Technology, University of New England, Armidale, NSW, Australia, **5** Division of Exercise Science and Sports Medicine, Department of Human Biology, Faculty of Health Sciences, the University of Cape Town and the Sports Science Institute of South Africa, Cape Town, South Africa, **6** School of Built Environment, Engineering and Computing, Leeds Beckett University, Leeds, United Kingdom

* J.Parmley8762@student.leedsbeckett.ac.uk

**Data Availability Statement:** Data are within the Supporting Information file.

**Funding:** The author(s) received no specific funding for this work.

## Abstract

This study investigated differences in external training load between microcycle lengths and its variation between microcycles, players, and head coaches. Commonly used external training load variables including total-, high-speed- (5–7 m·s$^{-1}$), and sprint-distance (> 7 m·s$^{-1}$) alongside combined high acceleration and deceleration distance (> 2 m·s$^{-2}$). Which were also expressed relative to time were collected using microtechnology within a repeated measures design from 54 male rugby league players from one Super League team over four seasons. 4337 individual observations across ninety-one separate microcycles and six individual microcycle lengths (5 to 10 day) were included. Linear mixed effects models established the differences in training load between microcycle-length and the variation between-microcycles, players and head coaches. The largest magnitude of difference in training load was seen when comparing 5-day with 9-day (ES = 0.31 to 0.53) and 10-day (ES = 0.19 to 0.66) microcycles. The greatest number of differences between microcycles were observed in high- (ES = 0.3 to 0.53) and sprint-speed (ES = 0.2 to 0.42) variables. Between-microcycle variability ranged between 11% to 35% dependent on training load variable. Training load also varied between players (5–65%) and head coaches (6–20%) with most variability existing within high-speed (19–43%) and sprinting (19–65%). Overall, differences in training load between microcycle lengths exist, likely due to manipulation of session duration. Furthermore, training load varies between microcycle, player and head coach.

## Introduction

Domestically, rugby league is played professionally in England and France in the Super League (SL) and in Australia and New Zealand in the National Rugby League (NRL). During a

**Competing interests:** The authors have declared that no competing interests exist.

competitive season, players undertake matches on a weekly basis meaning practitioners are tasked with finding a balance between training and recovery [1]. There is a general acceptance that an appropriate training load periodization between match-days is beneficial to the management of training outcomes [2, 3]. However, during the season the length of the training period between match days or microcycle is not consistent due to fixture schedules [4]. Understanding the effect of microcycle length on training load prescription allows greater insights into periodization strategies during the in-season period of professional rugby league.

Previous research in the NRL has reported significantly lower on-field training loads during shorter- (5–6 day [209 ± 63 AU]) than moderate- (7–8 day [235 ± 46 AU]) or longer- (9–10 day [242 ± 40 AU]) microcycles using session rating of perceived exertion (RPE) multiplied by the session duration (sRPE-TL) [1]. Furthermore, differences in on and off-field training load between 5-day (1309 ± 436 AU), 7-day (1704 ± 419) and 9-day (1862 ± 552 AU) micro-cycles have been reported [5]. Such differences were attributed to clubs prioritising recovery post-match over training frequency during shorter microcycles [1, 5]. Although, using solely RPE as a global measure of training load makes it difficult to understand how the specific training prescription (i.e., components of external load) differs across microcycles. In SL, differences in external training load have been reported across stages of the season (i.e., pre-, early-, mid-, and late-season) [6]. To date, the most longitudinal study assessing sRPE-TL by microcycle length was conducted across a single NRL season [1] which acknowledged some variation in training load within microcycle lengths. However, previous studies assessing differences in training load between microcycle lengths are limited in design due to the sole use of internal load, a lack of longitudinal repeated measures designs and discretisation of periods of training. This limits the ability to inform the prescription of specific microcycle lengths or understand their variability over time.

Training load variability is common within team sports with numerous contextual factors affecting subsequent weekly load including previous match outcome and phase of the season [7], along with variability between players occurring due to player's positional specific roles and their technical and physical capabilities [8]. It is logical that along with variability between players, training load data would also be influenced by the head coach's training philosophy. However, this has not been investigated within rugby league. In soccer, the head coach was shown to affect the external training load within specific drills [9]. In practice, an understanding of the variability of training load data is important as it aids practitioners in interpreting whether a change in between microcycle training load is meaningful. Moreover, understanding of the effect of head coach on the prescription of training load in rugby league would allow support staff to better aid the transition between coaches.

Therefore, the primary aim of this study was to quantify and compare commonly used external training load metrics between different between-match microcycle lengths. A secondary aim was to determine total weekly training load variation due to microcycle, individual and head coach.

## Methods

### Experimental approach to the problem

A repeated measures design was used to investigate external load of one SL team over four seasons (2017, 2018, 2019 and 2020). External training load was quantified using microtechnology during all "in-season" field-based training completed as part of the clubs usual training programme. Microcycles were defined as the number of days between matches. Ninety-one separate microcycles, with six different microcycle lengths were included in the analysis: 5-day (n = 14), 6-day (n = 15), 7-day (n = 33), 8-day (n = 18), 9-day (n = 7) and 10-day (n = 4).

## Subjects

A total of 4,337 observations from 54 male rugby league players, consisting of 31 forwards and 23 backs (age = 26.2 ± 4.7 years, height = 185.8 ± 7.8 cm, body mass = 97 ± 15.6 kg) (mean = 80 ± 55, range = 8 to 202 observations per player) and four head coaches were analysed in the study. Ethical approval was granted by the Leeds Beckett University ethics committee (approval number: 35556) and players provided written consent to participate in the study.

## Procedures

Each participant wore a micro-electrical mechanical system (MEMS) device with enabled Global Navigation Satellite System (GNSS) technology (Catapult S5, Catapult Innovations, Melbourne, Australia). These provide geospatial positioning at a 10 Hz sampling frequency encompassing both GPS (Global Positioning System) and GLONASS (Global Navigation Satellite System) satellites. It also contains an embedded 100Hz tri-axial accelerometer, gyroscope, and magnetometer. The device was worn in a company made vest which placed the unit tightly between the participants scapulae to reduce movement of the device. Each player wore the same unit in order to reduce the effects of inter device variation [10]. The mean ± SD number of satellites connected during data collection was 11.9 ± 0.12, whilst the horizontal dilution of precision (HDOP) was 0.72 ± 0.06 demonstrating acceptable quality of the signal [11]. Post-training data from the devices was calculated and processed through the manufacturer's proprietary software (Catapult Openfield, Catapult Sports, Melbourne Australia). The validity and reliability of these devices has previously been undertaken, demonstrating adequate reliability and validity to measure instantaneous speeds across multiple starting velocities (CV% = 2.0% to 5.3%) [12].

A number of locomotor-based metrics were used to measure external training load including total distance, high-speed running distance (5–7 $m \cdot s^{-1}$), and sprint distance ($> 7$ $m \cdot s^{-1}$). Thresholds were selected due to their frequent use within rugby league research [13]. Combined high acceleration and deceleration distance ($> 2$ $m \cdot s^{-2}$) was also selected for analysis. Several measures of training load, relative to time were also selected for analysis including average speed ($m \cdot min^{-1}$), high-speed running distance and high-acceleration and deceleration ($> 2$ $m \cdot s^{-2}$) distance ($m \cdot min^{-1}$). A minimum effort duration of 1 second was applied to reduce efforts caused by GPS error and random spikes in velocity [11]. All measures were aggregated to create microcycle total values for each metric.

## Statistical analyses

For descriptive purposes, total weekly training loads are presented as the mean ± standard deviation where appropriate. Total weekly loads were calculated by first summing the individual daily loads of players to create an overall daily load, then daily loads were summed to create a total microcycle load. Data analyses were carried out using mixed effects modelling with training load metrics (total distance, high-speed running distance (5–7 $m \cdot s^{-1}$), sprint distance ($> 7$ $m \cdot s^{-1}$), combined high acceleration and deceleration distance ($> 2$ $m \cdot s^{-2}$) average speed ($m \cdot min^{-1}$), high-speed running distance and high-acceleration and deceleration ($> 2$ $m \cdot s^{-2}$) distance ($m \cdot min^{-1}$)) as dependent variables. All dependent variables were log-transformed prior to analysis and subsequently back transformed to ensure the residuals were normally distributed [14]. Microcycle length was defined as the only fixed effect within the model. The random effects were individual players (player ID), the head coach at the time of data collection and the microcycle start date to account for between microcycle variation of external training load. Random effects remained constant throughout analysis. Cohen's d effect size (ES)

**Table 1. Description of microcycle training prescription for each microcycle length (mean ± SD).**

|  | Microcycle Length | | | | | |
|---|---|---|---|---|---|---|
|  | **5** | **6** | **7** | **8** | **9** | **10** |
| Microcycle Occurrence per Season (n) | 4 ± 2 | 4 ± 2 | 11 ± 2 | 6 ± 3 | 2 ± 1 | 1 ± 1 |
| Training Frequency (n) | 1.9 ± 0.5 | 2.1 ± 0.6 | 2.8 ± 0.4 | 2.8 ± 0.6 | 3.1 ± 1.1 | 2.8 ± 0.5 |
| Rest Frequency (n) | 3.1 ± 0.5 | 3.9 ± 0.6 | 4.2 ± 0.4 | 5.2 ± 0.6 | 4.9 ± 1.1 | 7.3 ± 0.5 |
| Percentage of Available Days utilised for Training (%) | 39 ± 9 | 34 ± 10 | 40 ± 6 | 35 ± 7 | 35 ± 12 | 28 ± 5 |
| Duration (mins) | 83 ± 29 | 89 ± 22 | 122 ± 26 | 128 ± 22 | 151 ± 71 | 132 ± 13 |

statistics [15] with 95% confidence intervals were estimated from the ratio between the mean difference to the pooled standard deviation for the fixed effect. ES were interpreted as < 0.20 = trivial, 0.20–0.59 = small, 0.60–1.19 = moderate, 1.20–2.00 = large, > 2.00 = very large. Null hypothesis testing was used to determine statistical significance (P < 0.05). Variability was calculated through the back-transformed standard deviation of the random effects expressed as a percentage. Statistical analysis was carried out in RStudio (version 1.3.1093) via the *lme4* package (version 1.1–26) [16].

## Results

Descriptions of microcycle length occurrence per season, training frequency, rest frequency, percentage of available days utilised for training and training duration are shown in Table 1.

Descriptive weekly training loads for each microcycle length are presented for commonly used training load metrics in Table 2.

Comparison of weekly training load during different microcycle lengths are presented in Fig 1 as forest plots of ES. Overall, the magnitude of difference for all investigated training load variables between microcycle lengths ranged from *trivial to moderate* (ES value range = 0.03 to 0.66).

Table 3 shows the variance of the model explained by each random effect for all weekly training load metrics (Microcycle-to-Microcycle = 11% to 35%, Player-to-Player = 5% to 65% and Coach-to-Coach = 6% to 20%).

## Discussion

The primary aim was to compare the difference in external training load variables across different between-match microcycle lengths in professional rugby league players over multiple seasons. The study also aimed to assess the variation in training load due to microcycle, individual and head coach. The present study generally found *trivial* to *moderate* differences in

**Table 2. Description of training load and intensity for each microcycle length (mean ± SD).**

|  | Microcycle Length (days) | | | | | |
|---|---|---|---|---|---|---|
|  | **5** | **6** | **7** | **8** | **9** | **10** |
| Total Distance (m) | 5276 ± 1843 | 6099 ± 1526 | 8509 ± 1622 | 9043 ± 1623 | 10090 ± 3340 | 9153 ± 703 |
| Average Speed (m·min$^{-1}$) | 67 ± 13 | 68 ± 6 | 71 ± 8 | 71 ± 5 | 72 ± 14 | 70 ± 4 |
| High-Speed Running Distance (m) | 284 ± 102 | 330 ± 135 | 488 ± 151 | 534 ± 151 | 775 ± 386 | 621 ± 81 |
| Relative High-Speed Running Distance (m·min$^{-1}$) | 3.8 ± 1.6 | 3.8 ± 1.0 | 4.1 ± 1.7 | 4.3 ± 1.1 | 5.6 ± 2.3 | 4.7 ± 0.7 |
| Sprint Distance (m) | 16 ± 16 | 19 ± 19 | 35 ± 25 | 36 ± 37 | 47 ± 20 | 43 ± 21 |
| Acceleration and Deceleration Distance (m) | 267 ± 115 | 318 ± 142 | 435 ± 94 | 466 ± 133 | 572 ± 197 | 470 ± 65 |
| Relative Acceleration and Deceleration Distance (m·min$^{-1}$) | 3.3 ± 0.7 | 3.5 ± 0.8 | 3.6 ± 0.6 | 3.5 ± 0.7 | 3.9 ± 0.7 | 3.5 ± 0.4 |

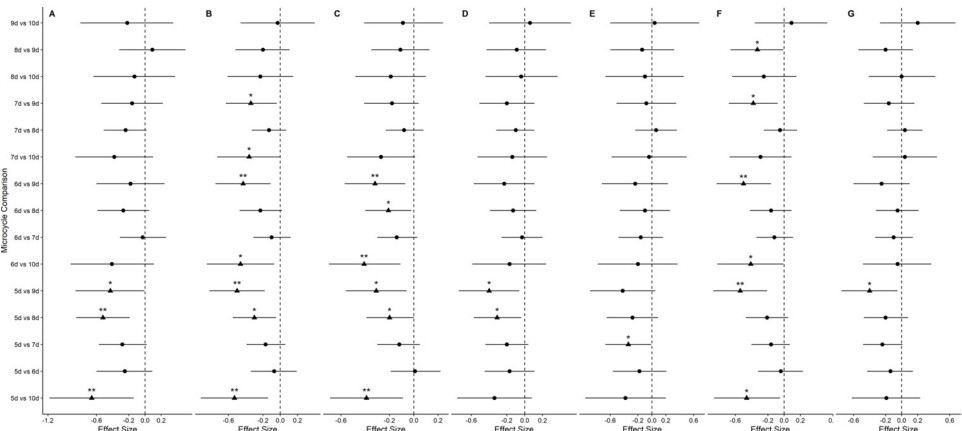

**Fig 1. Pairwise comparisons of training load by microcycle length.** (A) total distance (m), (B) high-speed running distance (m), (C) sprint distance (m), (D) high-acceleration and deceleration ($> 2$ m·s$^{-2}$) distance (m), (E) average speed (m·min$^{-1}$), (F) relative high-speed running distance (m·min$^{-1}$), (G) relative high-acceleration and deceleration ($> 2$ m·s$^{-2}$) distance (m·min$^{-1}$). Represented using Effect size 90% CI. Statistically significant comparisons are represented as ▲, while non-significant comparisons are represented as ●.* = $P \leq 0.05$, ** = $P \leq 0.01$, *** = $P \leq 0.001$.

training load between microcycle lengths. The microcycle comparisons with the largest magnitude of difference were 5-day and 9-day (ES = 0.31 to 0.53) and 5-day and 10-day (ES = 0.19 to 0.66) lengths, although these were *small* to *moderate* in magnitude. Moreover, the training load variables with the largest frequency of differences was observed in high- (7 comparisons, ES = 0.3 to 0.53) and sprint-speed (6 comparisons, ES = 0.2 to 0.41) variables. For a given microcycle length, training load varied 11% to 35% between weeks also differing between players (5–65%) and head coaches (6–20%) depending on the training load variable. Such variability data can provide practitioners with thresholds of typical variation in training load between microcycles, individuals and head coaches, allowing greater understanding of when systematic changes in training load have occurred.

## Microcycle comparisons

In the current study, 5-day microcycles showed smaller external loads (Table 2) for all variables when compared to longer microcycles ($> 7$-day) showing consistency with previous findings

**Table 3. The variance of the model explained by each random effect for all weekly training load metrics.** Expressed as a percentage derived from the back-transformed standard deviation of the random effects.

|  | Random Effect | | | |
|---|---|---|---|---|
|  | Between Microcycle Variability | Between Player Variability | Between Coach Variability | Residual Variability |
| Total Distance (m) | 15% | 5% | N/A | 29% |
| Average Speed (m·min$^{-1}$) | 11% | 5% | 10% | 16% |
| High-Speed Running Distance (m) | 33% | 43% | 19% | 86% |
| Relative High-Speed Running Distance (m·min$^{-1}$) | 22% | 27% | 20% | 50% |
| Sprint Distance (m) | 35% | 65% | 19% | 134% |
| Acceleration and Deceleration Distance (m) | 21% | 21% | 5% | 54% |
| Relative Acceleration and Deceleration Distance (m·min$^{-1}$) | 13% | 15% | 6% | 30% |

Nb coach to coach variability was not able to be calculated for total distance (m) but was maintained within the model for consistency with other models.

in rugby league [1, 5]. In order to manipulate the training load across a microcycle, practitioners can modify the frequency, duration and intensity of training across the week to achieve a desired training load. This reduction in load has previously been attributed to reduction in training frequency to prioritise recovery during shorter microcycles [1, 5]. Indicators of fatigue, such as muscle soreness and reductions in neuromuscular function are likely present for 48 hours following matches [17, 18] and are present regardless of microcycle length. Therefore, it is logical that such a training strategy would be implemented to ensure players are fully recovered for the next match with longer microcycles providing opportunities for greater training load prescription.

During longer microcycles (> 7-day), it appeared that coaches incorporated greater high-speed (ES = 0.23 to 0.53) and sprinting distance (ES = 0.2 to 0.41) than shorter microcycle lengths (< 7-day) (Fig 1). Given the longer time between matches, this could be due to a greater frequency of conditioning/interval-based training prescription rather than technical-tactical training, which has been found to elicit much greater high-speed running (> 5 m·s⁻¹) per min (82 m·min⁻¹ and 2.7 m·min⁻¹ respectively) [19, 20]. However, while the intensity and duration of high-speed and sprinting activity differed between longer and shorter length microcycles, average speed and relative acceleration and deceleration distance did not typically differ (Fig 1) showing consistency with previous research which used RPE as a measure of session intensity [1]. Therefore, it appears that unless there was a specific opportunity to prescribe additional high-speed and sprinting activity, practitioners did not tend to manipulate the intensity of training as a result of microcycle length.

In the present study, longer microcycles (> 7-day) exhibited greater mean training frequencies (mean = 2.8–2.9 training days) than shorter microcycle lengths (mean = 1.9–2.1 training days) (Table 1). However, during shorter microcycles it appears that training load was distributed over a similar proportion of available training days (35–39%) compared to longer microcycles (28–40%). Interestingly, 10-day microcycles had the lowest percentage of available days being utilised for training. As they occurred on average once per season this could have provided a logical opportunity to provide players with extended periods of rest during the season (Table 1). Additionally, descriptive data shown in Table 1 shows that rest frequency decreases with microcycle length, indicating that within this cohort recovery is not prioritised over training frequency contrary to suggestions within the previous research.

Collectively, the overall findings of the current study suggest a strategy to manipulate the duration of individual training sessions within microcycles rather than the intensity or frequency of training. More work is needed to understand whether reducing the training load but maintaining the training frequency across short duration microcycles is an optimal strategy to balancing training and recovery.

### Variability

The present study identified sources of variability (i.e microcycle, player and head coach) within external training load (Table 3). Our data shows between microcycle variability ranges between 11% and 35%, with average speed exhibiting the lowest between microcycle variation (11%), and sprint distance and high-speed running distance exhibiting the greatest amount of variability (35% and 33% respectively). The variation between microcycles may be explained by a myriad of contextual factors affecting training load including phase of the season [6], previous match result and quality of the next opponent [7]. Player-to-Player variability ranges from 5% in total distance and average speed, up to 65% in sprint distance (Table 3). Differences in position and the individual physical qualities (e.g., speed and strength) likely influence the between-player variation in training load seen during microcycles, as seen during match-

play [8, 21, 22]. Between head coach variability ranges from 5% to 20% with the lowest variation seen in combined acceleration and deceleration distance and highest variation seen in relative high-speed running distance. Such variation is most likely attributed to differences in tactical style between coaches with different match tactics translating into training, subsequently leading to variance in training load. Observations within the present study highlight those seen within other sports that a change in coaching personnel could affect the training load experienced by players [9].

### Limitations

Finally, there are limitations within the current study that should be acknowledged. Firstly, data was only collected from one club meaning that the findings may only be specific to the players and coaches within that club. A second limitation of the study is the sole use of external load measurements, the inclusion of an internal load variable would provide a more holistic view of training load and variances between players and head coaches [23]. The greatest limitation of the study is the lack of quantification of collision data. Rugby league is characterised by a combination of locomotion and repeated high intensity collisions and wrestling bouts [8]. Given the additional physiological cost of these actions, quantification of both the frequency and magnitudes of collision data is needed to provide rugby league practitioners more insight into training load experienced by players.

### Conclusion

The present study found small differences in load between microcycle lengths, with shorter microcycle lengths exhibiting lower training loads than longer microcycles. These differences appear to be attributed to changes in training duration rather than intensity and frequency of training. The greatest differences in external training load between microcycle lengths were for high speed and sprinting activity, which was greater during longer microcycles. This study also assessed and found variability in training load between microcycles, players, and the head coach finding that similar to previous research, the most variability exists within high speed and sprinting activity. Such data can be used by practitioners to understand whether a change in training load is systematic or as a result of natural variation based on the aforementioned factors.

### Practical applications

Practically, the microcycle comparison data within this study can be used to aid practitioners by informing training prescription and periodisation during different length microcycles, however the descriptive data in this study should not be considered as normative data as they are only specific to one club. The variability data presented in the present study can facilitate a practitioner's understanding of when systematic changes in external training load have occurred that is not due to the aforementioned factors. For example, to be confident a systematic change in high-speed running distance had occurred between head coaches, any change would need to exceed 15%. Therefore, in the current study, during a 7-day microcycle high-speed running would need to exceed 561m or be lower than 414m for this to occur.

### Supporting information

**S1 Data.**
(CSV)

## Author Contributions

**Conceptualization:** Tom Sawczuk.

**Supervision:** Ben Jones, Dan Weaving.

**Writing – original draft:** James Parmley.

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
