## [Decision Letter · Decision Letter 0]

2 Dec 2021

PONE-D-21-33345A four-season study quantifying the weekly external training loads during different between match microcycle lengths in professional rugby league.PLOS ONE

Dear Dr. Parmley,

Thank you for submitting your manuscript to PLOS ONE. After careful consideration, we feel that it has merit but does not fully meet PLOS ONE’s publication criteria as it currently stands. Therefore, we invite you to submit a revised version of the manuscript that addresses the points raised during the review process.

Based on the Reviewers' comments, we kindly invite to review the manuscript and send us a final version before we further process it.

We look forward to receiving your revised manuscript.

Kind regards,

Cristina Cortis, Ph.D.

Academic Editor

PLOS ONE

Journal Requirements:

2. PLOS requires an ORCID iD for the corresponding author in Editorial Manager on papers submitted after December 6th, 2016. Please ensure that you have an ORCID iD and that it is validated in Editorial Manager. To do this, go to ‘Update my Information’ (in the upper left-hand corner of the main menu), and click on the Fetch/Validate link next to the ORCID field. This will take you to the ORCID site and allow you to create a new iD or authenticate a pre-existing iD in Editorial Manager. Please see the following video for instructions on linking an ORCID iD to your Editorial Manager account: https://www.youtube.com/watch?v=_xcclfuvtxQ.

Reviewers' comments:

Reviewer's Responses to Questions

**Comments to the Author**

1. Is the manuscript technically sound, and do the data support the conclusions?

Reviewer #1: Yes

Reviewer #2: Yes

2. Has the statistical analysis been performed appropriately and rigorously? 

Reviewer #1: Yes

Reviewer #2: Yes

3. Have the authors made all data underlying the findings in their manuscript fully available?

Reviewer #1: Yes

Reviewer #2: Yes

4. Is the manuscript presented in an intelligible fashion and written in standard English?

Reviewer #1: Yes

Reviewer #2: Yes

5. Review Comments to the Author

Reviewer #1: Abstract

No indication about the type of external load parameters/instruments can be known by reading the abstract section. Therefore, it could be provided brief notes about this fundamental information to make more readable this independent section.

Introduction

Lines 58-62. Actually, the sole use of internal training load could determine limits as well as the sole use external loads. Therefore, for a complete training monitoring, both training loads should be considered. For this point, the following and recent study should be considered in the Introduction (and Discussion) section. Lupo C., Ungureanu A.N., Boccia G., Licciardi A., Rainoldi A., Brustio P.R. (2021). Internal training load monitoring, notational and time motion analyses, psychometric status, and neuromuscular responses in elite rugby union. International Journal of Sports Physiology and Performance. 5;16(3):421-428.

Discussion

Lines 212-218. Limitations. The missing consideration of at least one internal training load parameters should be considered as a limitation. The grade of divergences between training leaded by different head coaches as well as practiced by different players would be better controlled by means of internal training load parameters such as RPE and Well-being perception (Lupo et al., 2021).

Practical application

Even though these results have been originated from a single team collection data (as fairly reported among limitations), more general practical applications could be provide to make the article more useful for physical trainers and coaches.

Reviewer #2: General comments

The paper aimed to quantify and compare commonly used external training load metrics between different between-match microcycle lengths; and to determine total weekly training load variation due to microcycle, individual and head coach.

The paper is generally well written based on sound literature, the methods are clear, detailed and replicable, the results well-presented and discussed with respect to the literature.

When using abbreviations, make sure you explain it first and then use the correct abbreviation throughout the paper.

Introduction

The authors kept in mind that this section is a development of the hypotheses of the study leading to the purpose of the investigation. The most relevant literature has been included, and I think the section flows with no major concerns.

Materials and Methods

Overall, methods and procedures are clear, detailed and replicable.

Results and discussion

I think the results and discussion sections of the study are well presented and discussed with respect to the current literature.

Figure and tables

Figures and tables should be clear and stand on their own. Make sure you check this.

6. PLOS authors have the option to publish the peer review history of their article (what does this mean?). If published, this will include your full peer review and any attached files.

Reviewer #1: **Yes: **Corrado Lupo

Reviewer #2: No

---

## [Author Response · Author response to Decision Letter 0]

17 Dec 2021

Reviewer #1: 

Abstract

Comment: No indication about the type of external load parameters/instruments can be known by reading the abstract section. Therefore, it could be provided brief notes about this fundamental information to make more readable this independent section.

Response: Thank you for your comment, the external load variables used within the study have been included in the abstract which now reads

“Commonly used external training load variables including total-, high-speed- (5-7 m∙s-1), and sprint-distance (> 7 m∙s-1) alongside combined high acceleration and deceleration distance (> 2 m∙s-2). Which were also expressed relative to time were collected using microtechnology within a repeated measures design from 54 male rugby league players from one Super League team over four seasons. 4337 individual observations across ninety-one separate microcycles and six individual microcycle lengths (5 to 10 day) were included.”

Introduction

Comment: Lines 58-62. Actually, the sole use of internal training load could determine limits as well as the sole use external loads. Therefore, for a complete training monitoring, both training loads should be considered. For this point, the following and recent study should be considered in the Introduction (and Discussion) section. Lupo C., Ungureanu A.N., Boccia G., Licciardi A., Rainoldi A., Brustio P.R. (2021). Internal training load monitoring, notational and time motion analyses, psychometric status, and neuromuscular responses in elite rugby union. International Journal of Sports Physiology and Performance. 5;16(3):421-428.

Response: Thank you for the comment. We agree that for a holistic view of training load, a combination of both internal and external load measurements is important. However, we feel that this statement does not quite fit in the introduction whereby we state that previous studies assessing microcycle length have been limited by the sole use of internal training load as this does not provide an understanding of the training prescribed during different length microcycles i.e., components of the external load (Impellizierri, Marcora and Coutts, 2019).

Discussion

Comment: Lines 212-218. Limitations. The missing consideration of at least one internal training load parameters should be considered as a limitation. The grade of divergences between training leaded by different head coaches as well as practiced by different players would be better controlled by means of internal training load parameters such as RPE and Well-being perception (Lupo et al., 2021).

Response: Thank you for the comment, we agree that the inclusion of internal load variables may provide a more holistic view of training load thus has been listed as a limitation of the study. 

The section now reads “A second limitation of the study is the sole use of external load measurements, the inclusion of an internal load variable would provide a more holistic view of training load and variances between players and head coaches (Lupo et al., 2021)”

Practical application

Comment: Even though these results have been originated from a single team collection data (as fairly reported among limitations), more general practical applications could be provide to make the article more useful for physical trainers and coaches.

Response: Thank you for the comment. We have added in practical applications for the microcycle comparison data but have stressed that the data may only be specific to the players and coaches of one club.

Reviewer #2: General comments

Comment: The paper aimed to quantify and compare commonly used external training load metrics between different between-match microcycle lengths; and to determine total weekly training load variation due to microcycle, individual and head coach.

The paper is generally well written based on sound literature, the methods are clear, detailed and replicable, the results well-presented and discussed with respect to the literature.

When using abbreviations, make sure you explain it first and then use the correct abbreviation throughout the paper.

Introduction

The authors kept in mind that this section is a development of the hypotheses of the study leading to the purpose of the investigation. The most relevant literature has been included, and I think the section flows with no major concerns.

Materials and Methods

Overall, methods and procedures are clear, detailed and replicable.

Results and discussion

I think the results and discussion sections of the study are well presented and discussed with respect to the current literature.

Figure and tables

Figures and tables should be clear and stand on their own. Make sure you check this.

Response: Thank you for the comments they are greatly appreciated. We have ensured that all abbreviations have been explained previously in the paper namely “HDOP” in the methods section has now been explained as horizontal dilution of precision. We have also increased the resolution of figure 1 so now feel the figures and tables do stand on their own and the titles provide an adequate description to allow understanding.

---

## [Decision Letter · Decision Letter 1]

12 Jan 2022

A four-season study quantifying the weekly external training loads during different between match microcycle lengths in professional rugby league.

PONE-D-21-33345R1

Dear Dr. Parmley,

We’re pleased to inform you that your manuscript has been judged scientifically suitable for publication and will be formally accepted for publication once it meets all outstanding technical requirements.

Kind regards,

Cristina Cortis, Ph.D.

Academic Editor

PLOS ONE

Additional Editor Comments (optional):

All the comments have been successfully addressed and the paper can be now accepted for publication.

Reviewers' comments:

Reviewer's Responses to Questions

**Comments to the Author**

1. If the authors have adequately addressed your comments raised in a previous round of review and you feel that this manuscript is now acceptable for publication, you may indicate that here to bypass the “Comments to the Author” section, enter your conflict of interest statement in the “Confidential to Editor” section, and submit your "Accept" recommendation.

Reviewer #1: All comments have been addressed

Reviewer #2: All comments have been addressed

2. Is the manuscript technically sound, and do the data support the conclusions?

Reviewer #1: Yes

Reviewer #2: Yes

3. Has the statistical analysis been performed appropriately and rigorously? 

Reviewer #1: Yes

Reviewer #2: Yes

4. Have the authors made all data underlying the findings in their manuscript fully available?

Reviewer #1: Yes

Reviewer #2: Yes

5. Is the manuscript presented in an intelligible fashion and written in standard English?

Reviewer #1: Yes

Reviewer #2: Yes

6. Review Comments to the Author

Reviewer #1: (No Response)

Reviewer #2: I thank the authors for addressing all my comments and concerns. I have no more comments from my side.

7. PLOS authors have the option to publish the peer review history of their article (what does this mean?). If published, this will include your full peer review and any attached files.

Reviewer #1: **Yes: **Corrado Lupo

Reviewer #2: No

---

## [Editor Report · Acceptance letter]

21 Jan 2022

PONE-D-21-33345R1 

A four-season study quantifying the weekly external training loads during different between match microcycle lengths in professional rugby league. 

Dear Dr. Parmley:

I'm pleased to inform you that your manuscript has been deemed suitable for publication in PLOS ONE. Congratulations! Your manuscript is now with our production department. 

Kind regards, 

on behalf of

Prof. Dr. Cristina Cortis 

Academic Editor

PLOS ONE